# Asymmetric VAE for One-Step Video Super-Resolution Acceleration

## Abstract

Diffusion models have significant advantages in the field of real-world video super-resolution and have demonstrated strong performance in past research. In recent diffusion-based video super-resolution (VSR) models, the number of sampling steps has been reduced to just one, yet there remains significant room for further optimization in inference efficiency. In this paper, we propose FastVSR, which achieves substantial reductions in computational cost by implementing a high compression VAE (spatial compression ratio of 16, denoted as f16). We design the structure of the f16 VAE and introduce a stable training framework. We employ pixel shuffle and channel replication to achieve additional upsampling. Furthermore, we propose a lower-bound-guided training strategy, which introduces a simpler training objective as a lower bound for the VAE's performance. It makes the training process more stable and easier to converge. Experimental results show that FastVSR achieves speedups of 111.9 times compared to multi-step models and 3.92 times compared to existing one-step models.

## 1 Introduction

Video super-resolution (VSR) aims to reconstruct high-detail, high-fidelity videos from low-resolution inputs (Jo et al., 2018; Liang et al., 2024). VSR ensures temporal consistency by leveraging spatial structures within frames and temporal dependencies across frames. In this work, we focus on real-world video super-resolution (Real-VSR), where input videos are captured in natural environments and subject to unknown, time-varying degradations. Dynamic scenes introduce large, unpredictable motions and occlusions; camera systems contribute compression artifacts, sensor noise, rolling-shutter effects, and defocus blur; and degradation patterns may drift across devices and over time (Goodfellow et al., 2014; Lucas et al., 2019). Therefore, a practical Real-VSR system must simultaneously satisfy three requirements: (**i**) restore fine textures without generating false details, (**ii**) maintain temporal coherence to avoid flicker and identity drift, and (**iii**) meet strict latency and memory constraints for mobile capture, telepresence, surveillance, and streaming scenarios. Achieving all three remains highly challenging—alignment can be unstable under complex motion, per-frame enhancements may cause temporal inconsistency, and heavyweight models limit deployability—motivating solutions that optimize robustness and computational efficiency.

In this context, diffusion-based generative priors have proven to be particularly effective for VideoSR. Thanks to large-scale pretraining, they can generate rich textures and generalize well to degradations encountered in real-world scenarios (Blattmann et al., 2023; Zhang et al., 2023; Yang et al., 2025). Recent research has developed along two complementary directions. Multi-step diffusion VSR methods (Yang et al., 2024; Zhou et al., 2024; He et al., 2024; Xie et al., 2025) adapt image/video diffusion backbones and incorporate temporal information during denoising. Current approaches typically employ designs such as 3D convolutions, temporal layers, or optical-flow constraints to enforce temporal consistency, or leverage ControlNet and pretrained T2I/T2V models to provide strong priors. In parallel, one-step methods compress the entire denoising trajectory into a direct mapping from low resolution to high resolution (Chen et al., 2025; Sun et al., 2025). Representative strategies include consistency distillation (Song et al., 2023) from multi-step teacher models, flow matching (Lipman et al., 2022) with velocity supervision in latent space, and regression objectives combined with perceptual metrics; temporal coherence is maintained through clip-wise conditioning, shared noise schedules across frames, and lightweight temporal attention within a

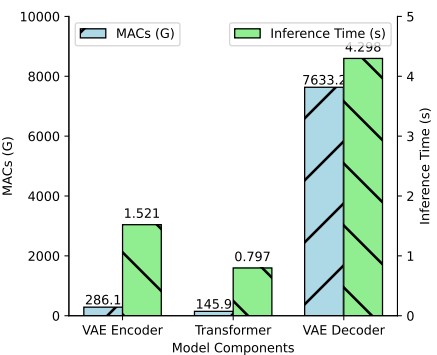 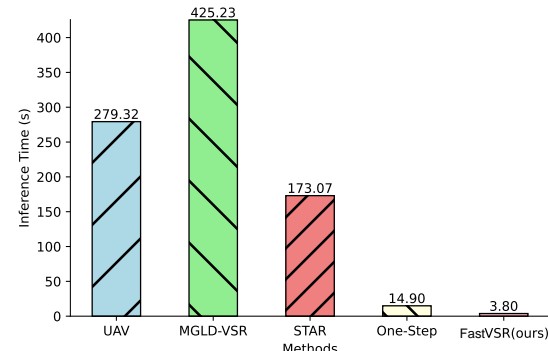

Figure 1: Left. The Multiply-Accumulate Operations (MACs) and inference time of the components in the CogVideoX model. Right. Comparison of inference times for different diffusion-based RealVSR models. The output size of video is $33 \times 720 \times 1280$.

single-pass generator. Together, these studies establish diffusion models as a strong foundation for VideoSR and motivate efficient, temporally aware designs.

Despite these advances, both approaches incur substantial inference overhead. Multi-step diffusion VSR requires dozens of solver evaluations per video clip; each forward pass traverses a high-parameter DiT or UNet combined with temporal modules and windowed context. As a result, runtime scales with the product of the number of steps, sequence length, and output resolution. Meanwhile, memory usage increases with cached key–value states and overlapping clips. Although one-step methods reduce latency by compressing the sampling trajectory into a single forward pass, their peak memory remains significant because the model must encode low-resolution frames into latent space and decode them at high resolution. As illustrated in Fig. 1, a simple computational analysis shows that once the denoising network is executed only once, the dominant cost shifts to the VAE codec: high-resolution decoding (and the accompanying encoding) accounts for most of the multiply–accumulate operations and activation footprint. This indicates that targeting the codec itself for compression and efficiency optimization can further accelerate inference and reduce memory usage, while preserving the benefits of diffusion priors for spatiotemporal restoration.

Building on this analysis, we propose FastVSR, a one-step framework centered on an asymmetric VAE to reduce the dominant codec cost and accelerate inference. Unlike conventional approaches that first interpolate low-resolution frames to the target size and then run the diffusion backbone at high resolution, we integrate part of the upsampling operation into the VAE codec. Specifically, we first encode the low-resolution video clip with an f8 VAE encoder, yielding a compact latent representation. Then we execute a one-step diffusion denoising through diffusion transformer. Subsequently, an f16 VAE decoder reconstructs the output at the target spatial scale. In effect, this constitutes indirect upsampling, governed by the scale ratio $r = f_{dec}/f_{enc}$. This asymmetric design reduces the number of spatial tokens processed by both the VAE and the DiT, lowers activation and cache memory usage, and confines the expensive spatial expansion to a single decoding pass, thereby providing significant end-to-end speedup and memory savings while preserving the advantages of diffusion priors for spatiotemporal restoration.

As shown in Fig. 1, **FastVSR** attains $\mathbf{111.9\times}$ speedup over multi-step diffusion baselines and $\mathbf{3.92\times}$ over prior one-step designs; at the same target resolution, peak memory usage is reduced by **46.3%**. In summary, our contributions are:

- We propose FastVSR, a one-step VideoSR framework built around an asymmetric VAE (f8 encoder / f16 decoder) that operates the diffusion transformer in a compact latent space and performs *indirect upsampling* at decode time.

- We design the architecture of the f16 VAE decoder and propose a lower-bound–guided training strategy to stably train the f16 VAE.

- Extensive experiments demonstrate $\mathbf{111.9\times/3.92\times}$ speedups and **46.3%** lower peak memory at comparable target resolutions, with high perceptual fidelity and temporal coherence across diverse real-world benchmarks.

## 2 RELATED WORK

### 2.1 DIFFUSION BASED REAL-VSR

Diffusion models have shown strong potential in Real-VSR by leveraging generative priors to restore high-quality textures from low-resolution input. These methods can be categorized into multi-step and one-step approaches. Multi-step methods adapt image/video diffusion backbones and perform iterative denoising with explicit temporal modeling: Upscale-A-Video (Zhou et al., 2024) exploits pretrained text-to-video priors with control branches to stabilize frame consistency; MGLD-VSR (Yang et al., 2024) couples optical-flow guidance with cross/deformable attention to align content across steps; VEnhancer (He et al., 2024) injects conditional video features via adapters/ControlNet within short-clip windows; and STAR (Xie et al., 2025) combines windowed context and temporal attention under DDIM/DPM-style samplers to handle complex motion. In contrast, one-step formulations compress the entire sampling trajectory into one forward pass to improve efficiency. Representative examples include DOVE (Chen et al., 2025) and DLoraL (Sun et al., 2025), which fine-tune pretrained video generators to perform one-step denoising, yielding competitive fidelity at markedly lower latency. Together, these lines of work establish diffusion as a compelling foundation for Real-VSR, trading off progressive stability against deployment-friendly speed while continuing to refine conditioning, alignment, and sampler design for real-world videos.

### 2.2 DEEP COMPRESSION VAE

Deep compression of variational autoencoders (VAEs) has proven effective for reducing the computational overhead of diffusion-based processing. By shrinking the latent space, it markedly lowers memory footprint and accelerates both encoding and decoding. DC-AE (Lu et al., 2025) increases the VAE compression ratio from f8 to f64 or higher, substantially cutting inference latency; however, it also requires expanding latent-channel capacity, which makes training the denoiser (e.g., DiT or U-Net) more difficult. Diffusion-4K (Zhang et al., 2025a;b) attains an f16 VAE by introducing additional upsampling operations and proposes a scale-consistent distillation loss. Moreover, hybrid approaches that combine compression strategies with diffusion and other deep generative models show strong potential for balancing performance and efficiency.

## 3 METHOD

In Section 3.1, we analyze the performance bottlenecks of current one-step diffusion models and motivate FastVSR. In Section 3.2, we present the design of the FastVSR architecture. In Section 3.3, we describe the training strategy for FastVSR.

### 3.1 MOTIVATION

One-step diffusion eliminates multi-iteration denoising but leaves intact the *pixel↔latent* coding cost. In Real-VSR, the VAE runs at the target high resolution (HR), whereas the one-step denoiser operates on a much smaller latent grid; empirically, run time and peak memory are therefore dominated by the codec—**especially the HR decoder**. Consequently, upsampling LR frames to HR before entering the VAE adds no information while disproportionately inflating the number of tokens and intermediate activations.

Let $V = THW$ be the HR spatiotemporal volume and $(s_t, s_s)$ the VAE temporal/spatial strides (so the latent volume is $V_{\text{lat}} = V/(s_t s_s^2)$). A compact scaling model is

$$\text{FLOPs} \approx \kappa_E V + \kappa_T \frac{V}{s_t s_s^2} + \kappa_D V, \tag{1}$$

$$\text{Act}_{\max} \approx \max\left\{ \mu_E V, \ \mu_T \frac{V}{s_t s_s^2}, \ \mu_D V \right\}, \tag{2}$$

with codec constants $\kappa_E, \kappa_D$ (encoder/decoder) and denoiser constant $\kappa_T$; typically $\kappa_D \gtrsim \kappa_E$, and $\kappa_D < \kappa_T < 50\kappa_D$. The decoder's dominance follows from the ratio

$$\frac{\text{FLOPs}_D}{\text{FLOPs}_T} \approx \frac{\kappa_D}{\kappa_T} s_t s_s^2 \gg 1 \quad (\text{e.g., } s_t{=}4, \ s_s{=}8 \Rightarrow s_t s_s^2{=}256). \tag{3}$$

Hence, after collapsing sampling to a single step, the **VAE—particularly HR decode—becomes the performance bottleneck**. This motivates a codec-centric design that reduces the VAE's HR

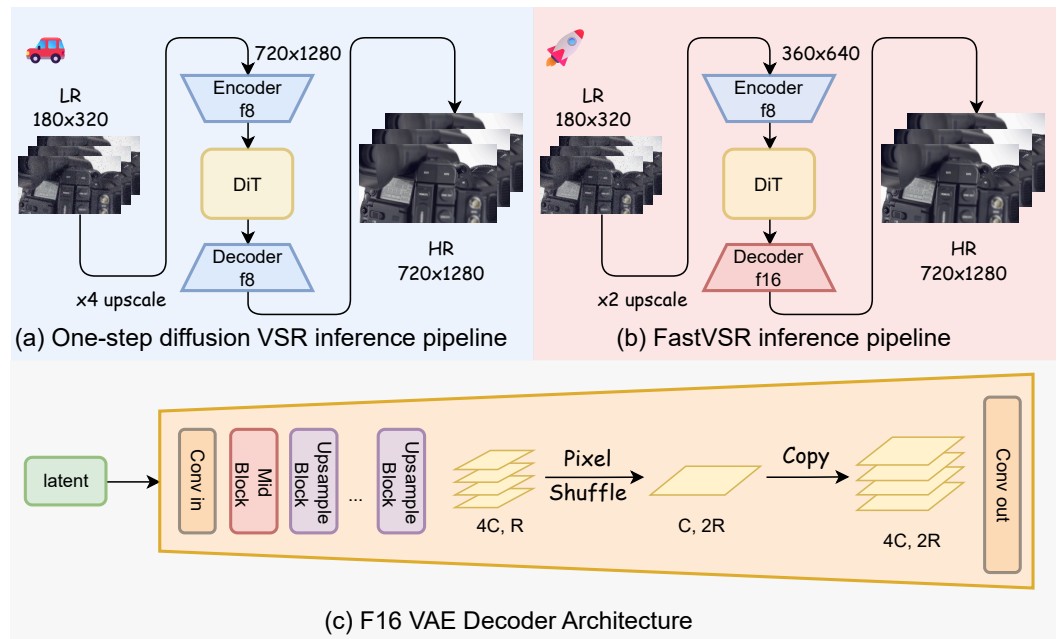

Figure 2: FastVSR Inference pipeline and details of f16 VAE Decoder.

workload (e.g., asymmetric, deep-compressed coding and deferring resolution expansion to a single efficient decode) to realize both speed and quality.

## 3.2 THE FASTVSR ARCHITECTURE

Guided by the above analysis, we adopt an asymmetric VAE with a f8 encoder and a f16 decoder. We keep the pretrained f8 encoder frozen and only fine-tunes an f16 decoder. The encoder therefore produces latents with the same statistics and geometry as the original model (identical channels and strides), so the one-step transformer can be reused without retraining. Resolution expansion is delegated to the decoder: it learns to map the unchanged latent grid to the target HR space via a higher decoding stride (effective scale governed by $r = f_{dec}/f_{enc}$), implementing indirect upsampling while preserving the latent space distribution. This separation preserves compatibility with strong pretrained transformers, stabilizes training (no latent drift), and concentrates learning capacity where the compute bottleneck lies—the HR decode path.

Figure 2(b) illustrates the overall design. Taking $4\times$ super-resolution as an example, the LR clip is first enlarged using simple $2\times$ interpolation upsampling and then fed into the one-step denoiser. FastVSR only requires training the f16 decoder, while the encoder and DiT remain frozen. During inference, all settings are consistent with the pretrained one-step diffusion model. This design ensures that the latent space distribution remains unchanged, avoiding the overhead of retraining the DiT, while ensuring plug-and-play compatibility of the f16 VAE.

We modify the VAE decoder head to realize the effective *f16* expansion. As shown in Fig. 2(c), a `PixelShuffle`(2) (Shi et al., 2016) is inserted immediately before the output layer to convert channels into spatial resolution, providing an additional $\times 2$ enlargement. After shuffling, we expand the feature tensor's channels (e.g., via duplication or a lightweight $1\times1$ projection), and then apply a final output convolution to produce the decoded HR image. To accommodate this topology change, the new output head is *randomly initialized*, while the remaining parts of the decoder are initialized from the pretrained model. Then we fine-tune the full parameters of decoder.

## 3.3 TRAINING STRATEGY: LOWER-BOUND–GUIDED F16 VAE

**Motivation.** Directly training an f16 decoder with MSE+perceptual(+GAN) in Real-VSR often leads to non-convergence and pseudo-textures. We replace adversarial supervision with an *optimiz-*

Figure 3: Lower-bound guided training strategy for F16 VAE decoder.

*able lower bound*: two VAEs provide stable, probabilistically grounded signals through their free energies (negative ELBO), guiding the f16-VAE toward the real data distribution.

**Dual VAEs and the optimizable lower bound.** We introduce two VAEs that play complementary roles over the same reconstruction $\hat{y}$: a frozen *reference VAE* $\mathcal{V}_{\mathrm{ref}}$ trained on real HR data, and a trainable *lower-bound VAE* $\mathcal{V}_{\mathrm{lb}}$ fine-tuned on the current generator outputs. For a generic VAE $\mathcal{V}_{\bullet}$ with encoder $q_{\bullet}(z|y)$, decoder $p_{\bullet}(y|z)$, and prior $p(z)$, its free energy (negative ELBO) is

$$F_{\bullet}(y) \;=\; \mathbb{E}_{q_{\bullet}(z|y)}\big[-\log p_{\bullet}(y|z)\big] \;+\; \mathrm{KL}\big(q_{\bullet}(z|y)\,\|\,p(z)\big), \qquad (4)$$

and satisfies the variational inequality $\log p_{\bullet}(y) \geq -F_{\bullet}(y)$. Where $\bullet$ denotes a generic VAE, which can represent either the reference VAE $\mathcal{V}_{\mathrm{ref}}$ or the lower-bound VAE $\mathcal{V}_{\mathrm{lb}}$. Applying this to $\mathcal{V}_{\mathrm{ref}}$ and $\mathcal{V}_{\mathrm{lb}}$ yields a *likelihood-ratio lower bound* for any $y$:

$$\log \frac{p_{\mathrm{ref}}(y)}{p_{\mathrm{lb}}(y)} \;\geq\; -F_{\mathrm{ref}}(y) \;+\; F_{\mathrm{lb}}(y). \qquad (5)$$

Evaluated on reconstructions $y = \hat{y}$, the right-hand side is computable and differentiable w.r.t. $\hat{y}$, so we define the surrogate objective

$$\mathcal{L}_{\mathrm{bound}}(\hat{y}) \;=\; F_{\mathrm{ref}}(\hat{y}) \;-\; F_{\mathrm{lb}}(\hat{y}), \qquad (6)$$

and *minimize* $\mathcal{L}_{\mathrm{bound}}$ to increase the lower bound on $\log \frac{p_{\mathrm{ref}}}{p_{\mathrm{lb}}}$, pushing $\hat{y}$ toward regions that the reference model assigns higher probability than the lower-bound model.

To see why this guides the f16-VAE, let $q_\theta(y)$ denote the generator-induced distribution over reconstructions for parameters $\theta$. Taking expectations gives

$$\underbrace{\mathbb{E}_{q_\theta}\big[\log p_{\mathrm{ref}}(y)\big]}_{\text{match to data via } \mathcal{V}_{\mathrm{ref}}} - \underbrace{\mathbb{E}_{q_\theta}\big[\log p_{\mathrm{lb}}(y)\big]}_{\text{match to generator via } \mathcal{V}_{\mathrm{lb}}} \;\geq\; -\mathbb{E}_{q_\theta}\big[F_{\mathrm{ref}}(y) - F_{\mathrm{lb}}(y)\big]. \qquad (7)$$

The lower-bound VAE $\mathcal{V}_{\mathrm{lb}}$ is trained on $q_\theta$ to *maximize* its ELBO (equivalently, *minimize* $F_{\mathrm{lb}}$), so $p_{\mathrm{lb}}$ tracks $q_\theta$ as a variational proxy. In the idealized limit $p_{\mathrm{lb}} \approx q_\theta$,

$$\mathbb{E}_{q_\theta}\big[\log p_{\mathrm{ref}}(y)\big] \;-\; \mathbb{E}_{q_\theta}\big[\log q_\theta(y)\big] \;=\; -\mathrm{KL}\big(q_\theta \,\|\, p_{\mathrm{ref}}\big) \;+\; H(q_\theta), \qquad (8)$$

so maximizing the (tractable) lower bound effectively reduces the reverse KL between the generator and the data up to an entropy term. Replacing $\log q_\theta$ with the trainable surrogate $\log p_{\mathrm{lb}}$ and *tightening* its ELBO by updating $\mathcal{V}_{\mathrm{lb}}$ yields a practical lower-bound optimization.

In practice, we couple this bound with standard pixel/perceptual terms to anchor fidelity, while gradients from $F_{\mathrm{ref}}$ and $F_{\mathrm{lb}}$ flow only through $\hat{y}$ (stop-grad on $\mathcal{V}_{\mathrm{ref}}$ and $\mathcal{V}_{\mathrm{lb}}$ during the generator update). The resulting signal $\nabla_{\hat{y}} F_{\mathrm{ref}} - \nabla_{\hat{y}} F_{\mathrm{lb}}$ behaves like an energy-based, margin-style critic: the reference branch provides a "move toward real" direction, and the lower-bound branch—trained on current outputs—provides a calibrated baseline, yielding stable, non-adversarial supervision that improves fidelity and suppresses pseudo-textures.

**Losses.** Let $\hat{y}$ be the f16-VAE reconstruction and $y^{\star}$ the HR ground truth. Define the free energy (negative ELBO) for a VAE $\bullet \in \{\mathrm{ref}, \mathrm{lb}\}$ as

$$F_{\bullet}(y) \;=\; \mathbb{E}_{q_{\bullet}(z|y)}\big[-\log p_{\bullet}(y|z)\big] \;+\; \mathrm{KL}\big(q_{\bullet}(z|y)\,\|\,p(z)\big). \tag{9}$$

The reconstruction loss and the lower-bound contrastive term are

$$\mathcal{L}_{\mathrm{rec}} = \lambda_{\mathrm{MSE}}\|\hat{y} - y^{\star}\|_2^2 + \lambda_{\mathrm{perc}}\big\|\Phi(\hat{y}) - \Phi(y^{\star})\big\|_2^2, \qquad \mathcal{L}_{\mathrm{bound}} = F_{\mathrm{ref}}(\hat{y}) - F_{\mathrm{lb}}(\hat{y}), \tag{10}$$

and the f16-VAE objective is

$$\mathcal{L}_{\mathrm{f16}} = \mathcal{L}_{\mathrm{rec}} + \lambda_b\,\mathcal{L}_{\mathrm{bound}} + \lambda_{\mathrm{reg}}\,\mathcal{R}(\hat{y}), \tag{11}$$

where $\Phi(\cdot)$ extracts perceptual features and $\mathcal{R}$ is an optional light regularizer (e.g., TV or color consistency).

**Stage A: Update F16-VAE.** Let $\theta$ denote f16-VAE parameters, and $\psi, \phi$ the parameters of $\mathcal{V}_{\mathrm{ref}}$ and $\mathcal{V}_{\mathrm{lb}}$, respectively. With $\psi, \phi$ frozen, we solve

$$\min_{\theta}\ \mathcal{L}_{\mathrm{f16}}(\theta), \tag{12}$$

and treat $F_{\mathrm{ref}}(\hat{y})$ and $F_{\mathrm{lb}}(\hat{y})$ as functions of $\hat{y}$ only (stop-grad on $\psi, \phi$). By the chain rule,

$$\nabla_{\theta}\mathcal{L}_{\mathrm{f16}} = \left(\frac{\partial \hat{y}}{\partial \theta}\right)^{\top}\big[\nabla_{\hat{y}}\mathcal{L}_{\mathrm{rec}} + \lambda_b\big(\nabla_{\hat{y}}F_{\mathrm{ref}}(\hat{y}) - \nabla_{\hat{y}}F_{\mathrm{lb}}(\hat{y})\big) + \lambda_{\mathrm{reg}}\nabla_{\hat{y}}\mathcal{R}(\hat{y})\big]. \tag{13}$$

Here $\nabla_{\hat{y}}F_{\mathrm{ref}}$ steers reconstructions toward regions of higher reference likelihood, while $-\nabla_{\hat{y}}F_{\mathrm{lb}}$ provides a complementary contrastive direction.

**Stage B: Update Lower-Bound VAE.** With $\theta, \psi$ frozen and $\hat{y}$ treated as data, the lower-bound VAE is updated via the standard VAE objective

$$\min_{\phi}\ \mathcal{L}_{\mathrm{lb}}(\phi) = \mathbb{E}_{q_{\phi}(z|\hat{y})}\big[-\log p_{\phi}(\hat{y}|z)\big] + \beta\,\mathrm{KL}\big(q_{\phi}(z|\hat{y})\,\|\,p(z)\big), \tag{14}$$

where $\beta \geq 1$ (a $\beta$-VAE style coefficient) can sharpen the bound. Alternating Stage A and Stage B tightens the lower bound and supplies a stable, non-adversarial learning signal that improves fidelity and suppresses pseudo-textures.

# 4 EXPERIMENTS

## 4.1 EXPERIMENTAL SETTINGS

**Datasets.** The training data consist of both video and image datasets. For videos, we use REDS (Nah et al., 2019), which contains a total of 239 high-quality sequences. For images, we use LS-DIR (Li et al., 2023), consisting of 85k texture-rich, high-resolution images. For evaluation, we adopt both synthetic and real-world benchmarks. The synthetic sets include UDM10 (Tao et al., 2017), SPMCS (Yi et al., 2019), and YouHQ40 (Zhou et al., 2024), generated with the RealBasicVSR (Chan et al., 2022) degradation pipeline. The real-world sets are RealVSR (Yang et al., 2021), MVSR4x (Wang et al., 2023b), and VideoLQ (Chan et al., 2022); RealVSR and MVSR4x provide smartphone-captured LQ–HQ pairs, while VideoLQ consists of Internet videos without HQ references. All experiments are conducted at a $\times 4$ upscaling factor.

**Evaluation Metrics.** We evaluate performance using a suite of image- and video-quality metrics. For IQA, we report two fidelity measures: PSNR and SSIM (Wang et al., 2004), and four perceptual metrics: LPIPS (Zhang et al., 2018), DISTS (Ding et al., 2020), MUSIQ (Ke et al., 2021) and CLIP-IQA (Wang et al., 2023a). For VQA, we assess overall video quality with DOVER (Wu et al., 2023). To quantify temporal consistency, we compute $E_{warp}^{*}$ (i.e., $E_{warp} \times 10^{-3}$) (Lai et al., 2018). Together, these metrics provide a comprehensive evaluation of video quality.

**Implementation Details.** Our FastVSR is based on the text-to-video diffusion model CogVideoX1.5 (Yang et al., 2025). The VAE encoder and the transformer are not further trained; they are directly initialized from pretrained one-step model. We use an empty text prompt that is pre-encoded to reduce inference overhead. Training proceeds in two stages: first, we train on the video dataset for 20k iterations with a learning rate of $5 \times 10^{-5}$; then, we switch to a mixed image–video dataset for 100k iterations with a learning rate of $5 \times 10^{-6}$. The resolutions are 256 px and 512 px for the two stages, respectively. In both stages, we use the AdamW optimizer (Loshchilov et al., 2018) with $\beta_1 = 0.9$, $\beta_2 = 0.95$, and $\beta_3 = 0.98$.

Table 1: Quantitative comparison with state-of-the-art methods. The best and second best results are colored with red and blue, respectively.

| Dataset | Metric | RealESRGAN (Wang et al., 2021) | RealBasicVSR (Chan et al., 2022) | Upscale-A-Video (Zhou et al., 2024) | MGLD-VSR (Yang et al., 2024) | VEnhancer (He et al., 2024) | STAR (Xie et al., 2025) | SeedVR (Wang et al., 2025) | FastVSR (ours) |
|---|---|---|---|---|---|---|---|---|---|
| UDM10 | PSNR ↑ | 24.04 | 24.13 | 21.72 | 24.23 | 21.32 | 23.47 | 23.39 | 24.36 |
| | SSIM ↑ | 0.7107 | 0.6801 | 0.5913 | 0.6957 | 0.6811 | 0.6804 | 0.6843 | 0.7184 |
| | LPIPS ↓ | 0.3877 | 0.3908 | 0.4116 | 0.3272 | 0.4344 | 0.4242 | 0.3583 | 0.3496 |
| | DISTS ↓ | 0.2184 | 0.2067 | 0.2230 | 0.1677 | 0.2310 | 0.2156 | 0.1339 | 0.1628 |
| | CLIP-IQA ↑ | 0.4189 | 0.3494 | 0.4697 | 0.4557 | 0.2852 | 0.2417 | 0.3145 | 0.5947 |
| | MUSIQ ↑ | 55.67 | 59.06 | 59.91 | 60.55 | 37.25 | 41.98 | 53.62 | 58.16 |
| | DOVER ↑ | 0.7060 | 0.7564 | 0.7291 | 0.7264 | 0.4576 | 0.4830 | 0.6889 | 0.7638 |
| | $E^*_{warp}$ ↓ | 4.83 | 3.10 | 3.97 | 3.59 | 1.03 | 2.08 | 3.24 | 1.70 |
| SPMCS | PSNR ↑ | 21.22 | 22.17 | 18.81 | 22.39 | 18.58 | 21.24 | 21.22 | 22.48 |
| | SSIM ↑ | 0.5613 | 0.5638 | 0.4113 | 0.5896 | 0.4850 | 0.5441 | 0.5672 | 0.6020 |
| | LPIPS ↓ | 0.3721 | 0.3662 | 0.4468 | 0.3263 | 0.5358 | 0.5257 | 0.3448 | 0.3196 |
| | DISTS ↓ | 0.2220 | 0.2164 | 0.2452 | 0.1960 | 0.2669 | 0.2872 | 0.1611 | 0.1882 |
| | CLIP-IQA ↑ | 0.5238 | 0.3513 | 0.5248 | 0.4348 | 0.3188 | 0.2646 | 0.3945 | 0.6204 |
| | MUSIQ ↑ | 66.63 | 66.87 | 69.55 | 65.56 | 42.71 | 36.66 | 62.59 | 69.18 |
| | DOVER ↑ | 0.7490 | 0.6753 | 0.7171 | 0.6754 | 0.4284 | 0.3204 | 0.6576 | 0.7525 |
| | $E^*_{warp}$ ↓ | 5.61 | 1.88 | 4.22 | 1.68 | 1.19 | 1.01 | 1.72 | 1.27 |
| YouHQ40 | PSNR ↑ | 22.82 | 22.39 | 19.62 | 23.17 | 19.78 | 22.64 | 21.94 | 22.85 |
| | SSIM ↑ | 0.6337 | 0.5895 | 0.4824 | 0.6194 | 0.5911 | 0.6323 | 0.5914 | 0.6614 |
| | LPIPS ↓ | 0.3571 | 0.4091 | 0.4268 | 0.3608 | 0.4742 | 0.4600 | 0.3474 | 0.3513 |
| | DISTS ↓ | 0.1790 | 0.1933 | 0.2012 | 0.1685 | 0.2140 | 0.2287 | 0.1084 | 0.1491 |
| | CLIP-IQA ↑ | 0.4704 | 0.3964 | 0.5258 | 0.4657 | 0.3309 | 0.2739 | 0.4123 | 0.5888 |
| | MUSIQ ↑ | 60.37 | 65.30 | 67.75 | 62.10 | 59.69 | 34.86 | 60.77 | 65.38 |
| | DOVER ↑ | 0.8572 | 0.7636 | 0.8596 | 0.8446 | 0.6957 | 0.5594 | 0.8492 | 0.8558 |
| | $E^*_{warp}$ ↓ | 5.91 | 3.08 | 6.84 | 3.45 | 0.95 | 2.21 | 3.43 | 1.90 |
| RealVSR | PSNR ↑ | 20.85 | 22.12 | 20.29 | 22.02 | 15.75 | 17.43 | 20.14 | 22.20 |
| | SSIM ↑ | 0.7105 | 0.7163 | 0.5945 | 0.6774 | 0.4002 | 0.5215 | 0.6738 | 0.7230 |
| | LPIPS ↓ | 0.2016 | 0.1870 | 0.2671 | 0.2182 | 0.3784 | 0.2943 | 0.2466 | 0.1830 |
| | DISTS ↓ | 0.1279 | 0.0983 | 0.1425 | 0.1169 | 0.1688 | 0.1599 | 0.1185 | 0.0965 |
| | CLIP-IQA ↑ | 0.7472 | 0.2905 | 0.4855 | 0.4510 | 0.3880 | 0.3641 | 0.2996 | 0.5206 |
| | MUSIQ ↑ | 72.43 | 70.73 | 71.13 | 70.69 | 72.27 | 70.23 | 61.24 | 73.25 |
| | DOVER ↑ | 0.7542 | 0.7636 | 0.7114 | 0.7508 | 0.7637 | 0.7051 | 0.6778 | 0.7810 |
| | $E^*_{warp}$ ↓ | 6.32 | 4.45 | 6.25 | 3.16 | 5.15 | 9.88 | 3.72 | 3.40 |
| MVSR4x | PSNR ↑ | 22.47 | 21.80 | 20.42 | 22.77 | 20.50 | 22.42 | 21.54 | 22.55 |
| | SSIM ↑ | 0.7412 | 0.7045 | 0.6117 | 0.7418 | 0.7117 | 0.7421 | 0.6869 | 0.7480 |
| | LPIPS ↓ | 0.4534 | 0.4235 | 0.4717 | 0.3568 | 0.4471 | 0.4311 | 0.4944 | 0.3430 |
| | DISTS ↓ | 0.3021 | 0.2498 | 0.2673 | 0.2245 | 0.2800 | 0.2714 | 0.2229 | 0.2291 |
| | CLIP-IQA ↑ | 0.4396 | 0.4118 | 0.6106 | 0.3769 | 0.3104 | 0.2674 | 0.2371 | 0.6058 |
| | MUSIQ ↑ | 37.80 | 62.96 | 69.80 | 53.46 | 37.34 | 32.24 | 42.56 | 69.91 |
| | DOVER ↑ | 0.2111 | 0.6846 | 0.7221 | 0.6214 | 0.3164 | 0.2137 | 0.3548 | 0.6970 |
| | $E^*_{warp}$ ↓ | 1.64 | 1.69 | 5.10 | 1.55 | 0.62 | 0.61 | 2.73 | 1.02 |
| VideoLQ | CLIP-IQA ↑ | 0.3617 | 0.3433 | 0.4132 | 0.3465 | 0.3031 | 0.2652 | 0.2359 | 0.4513 |
| | MUSIQ ↑ | 49.84 | 55.62 | 55.04 | 51.00 | 42.35 | 39.66 | 39.10 | 53.11 |
| | DOVER ↑ | 0.7310 | 0.7388 | 0.7370 | 0.7421 | 0.6912 | 0.7080 | 0.6799 | 0.7432 |
| | $E^*_{warp}$ ↓ | 7.58 | 5.97 | 13.47 | 6.79 | 6.50 | 5.96 | 8.34 | 6.92 |

## 4.2 COMPARISON WITH STATE-OF-THE-ART METHODS

We compare FastVSR with state-of-the-art image and video super-resolution methods in the real-world video super-resolution (Real-VSR) setting, including RealESRGAN (Wang et al., 2021), RealBasicVSR (Chan et al., 2022), Upscale-A-Video (Zhou et al., 2024), MGLD-VSR (Yang et al., 2024), VEnhancer (He et al., 2024), STAR (Xie et al., 2025), and SeedVR (Wang et al., 2025).

**Quantitative Results.** We present a comprehensive comparison in Table 1, where we evaluate FastVSR across a variety of benchmarks on six datasets. Under the fixed ×4 upscaling setting, our method consistently demonstrates strong performance while maintaining significant efficiency advantages. Specifically, in terms of perception-driven metrics, FastVSR excels: LPIPS and DISTS values decrease (which is desirable, as lower values indicate better performance), and CLIPIQA achieves its best performance on several datasets, suggesting that our codec design successfully preserves high-frequency details without introducing excessive sharpening. Furthermore, the video-level quality remains consistently high: FasterVQA and DOVER both show optimal performance on the majority of datasets. Temporal consistency, as measured by the flow-warp error $E^*_{warp}$, performs well across various datasets. It reflects the robustness and accuracy of inter-frame reconstruction. These comprehensive results collectively demonstrate that FastVSR strikes an effective and favorable balance between high-quality output and computational efficiency.

**Qualitative Results.** Figure 4 visualizes representative crops from challenging regimes—texture-rich regions, heavily degraded scenes, and text overlays. In texture-rich areas (e.g., foliage, fabrics, building facades), FastVSR reconstructs fine patterns with clear edge and real details, while avoiding hallucinated textures. Under severe degradations (compression, noise, motion blur), it restores salient structures and suppresses blockiness and zippering, yielding visually coherent content. For text, FastVSR preserves stroke sharpness and character geometry without distortions or mis-generation, which is crucial for readability. Temporal inspection of consecutive frames shows reduced flicker in repetitive textures and stable object contours, consistent with the quantitative $E^*_{warp}$ gains. Additional examples, zoom-in views, and failure cases (e.g., very fast nonrigid motion) are provided in the supplementary material.

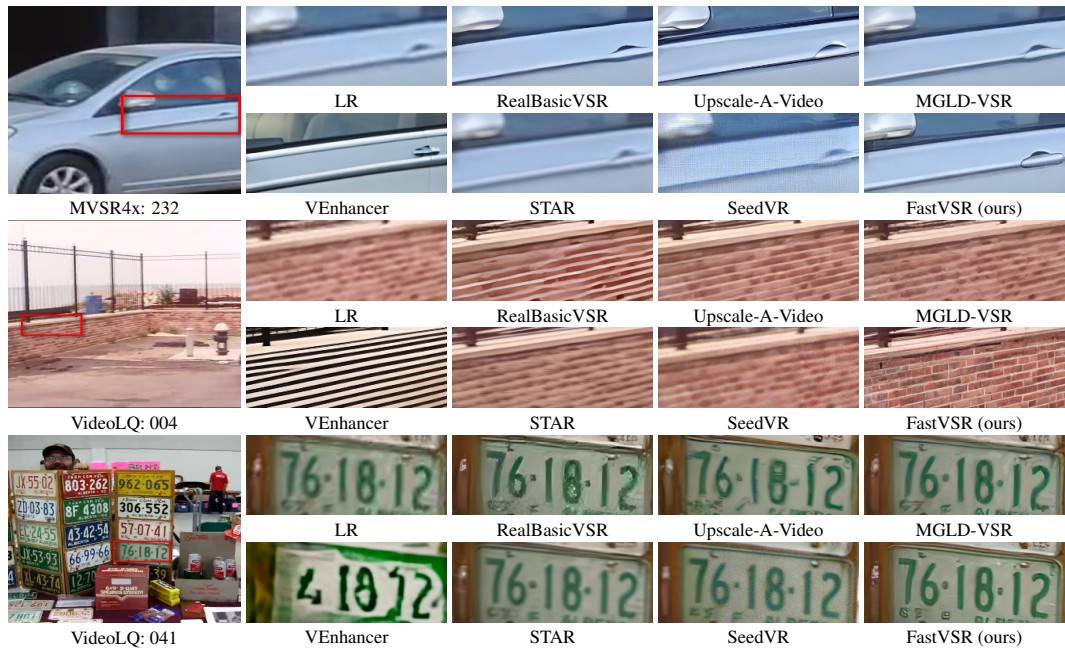

Figure 4: Visual comparison on real-world datasets (MVSR4x (Wang et al., 2023b) and VideoLQ (Chan et al., 2022)). The videos in VideoLQ are sourced from the Internet without high-resolution (HQ) references.

Table 2: Efficiency comparison across diffusion-based Real-VSR methods.

| Method | Sampling Steps | E2E Latency (s) | MACs(T) | Peak Memory (GB) |
|---|---|---|---|---|
| Upscale-A-Video | 30 | 279.32 | 9,084 | 14.87 |
| MGLD-VSR | 50 | 425.23 | 8,528 | 19.64 |
| VEnhancer | 15 | 121.27 | 5,273 | 11.71 |
| STAR | 15 | 173.07 | 4,281 | 18.06 |
| f8 VAE one-step VSR | 1 | 14.90 | 504.8 | 60.78 |
| FastVSR (ours) | 1 | 3.80 | 125.7 | 32.59 |

**Inference Efficiency.** As shown in Table 2, we benchmark efficiency across four dimensions: sampling steps, end-to-end (E2E) latency, compute (MACs), and peak memory, under a fixed $\times 4$ upscaling setting. FastVSR employs a one-step denoising, matching the one-step baselines, and contrasting sharply with multi-step diffusion methods, which typically require $15 \sim 50$ steps. E2E latency is reported as *image → VAE encoder → transformer → VAE decoder → image*. The video size is $33 \times 720 \times 1280$. In this setup, FastVSR significantly accelerates inference, achieving $111.9\times$ speedup over multi-step diffusion and $3.92\times$ over one-step diffusion. In terms of compute, we measure Multiply-Accumulate Operations (MACs) per clip: FastVSR primarily reduces total MACs by decreasing the input size for both the VAE and denoiser. At the same target resolution, peak memory is also reduced compared to one-step baselines. This is due to the asymmetric codec design, which defers spatial expansion to a single decoding pass and minimizes high-resolution activations. Taken together, these results establish FastVSR as both the fastest and most memory-efficient method among the diffusion-based Real-VSR approaches evaluated.

### 4.3 ABLATION STUDY

**Upsample method.** We compared different upsampling methods to evaluate their impact on the performance of the f16 VAE model. Specifically, we compared PixelShuffle with nearest, bilinear, and bicubic upsampling methods. As shown in Table 3a, while more complex upsampling methods can improve image quality, they come at the cost of reduced inference efficiency. Furthermore, Fig. 5 illustrates the pseudo-textures introduced by different upsampling methods. PixelShuffle strikes the best balance between image quality and inference speed.

**Training Strategy.** We compare our proposed training strategy with traditional methods such as vanilla training strategy (MSE + Perceptual loss + GAN) and knowledge distillation. The vanilla

Table 3: Ablation study on the effects of upsampling methods and training strategies. (a) Different upsampling methods include nearest, bilinear, bicubic, and pixel shuffle. (b) Comparison of different training strategies: vanilla (MSE + Perceptual loss + GAN), knowledge distillation (KD), and lower-bound guided (LBG, ours). All experiments are conducted on the UDM10 dataset.

(a) Ablation on upsampling method

| Upsampling Method | nearest | bilinear | bicubic | pixel shuffle |
|---|---|---|---|---|
| PSNR ↑ | 22.80 | 23.10 | **24.48** | 24.36 |
| LPIPS ↓ | 0.3701 | 0.3578 | 0.3522 | **0.3496** |
| MUSIQ ↑ | 52.28 | 55.12 | 56.93 | **58.16** |
| CLIP-IQA ↑ | 0.4600 | 0.5085 | 0.5107 | **0.5947** |
| DOVER ↑ | 0.7112 | 0.7503 | **0.7809** | 0.7638 |
| Inference time / s ↓ | **3.799** | 3.806 | 3.833 | 3.802 |

(b) Ablation on training strategy.

| Training Strategy | vanilla | KD | LBG(ours) |
|---|---|---|---|
| PSNR ↑ | 21.20 | 23.25 | **24.36** |
| SSIM ↑ | 0.5800 | 0.6990 | **0.7184** |
| LPIPS ↓ | 0.3661 | **0.3105** | 0.3496 |
| MUSIQ ↑ | 45.83 | 55.38 | **58.16** |
| CLIP-IQA ↑ | 0.4950 | 0.5107 | **0.5947** |
| DOVER ↑ | 0.7100 | 0.7444 | **0.7638** |

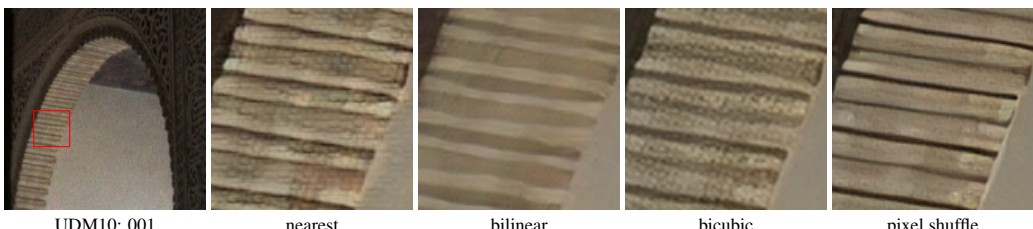

| UDM10: 001 | nearest | bilinear | bicubic | pixel shuffle |

Figure 5: Comparison of different upsample methods used in f16 VAE decoder.

training approach is commonly used for VAE training, where the loss includes pixel-wise MSE for fidelity, perceptual loss for high-level feature preservation, and a GAN loss for generating realistic textures. While this method performs well on f8 VAE, it struggles with high compression ratio VAE, resulting in slower convergence and the introduction of pseudo-textures. On the other hand, knowledge distillation, which involves training a student model by minimizing the divergence between its output and the output of a teacher model, is another widely used approach. While distillation can transfer knowledge from a powerful teacher, it still faces challenges related to training stability and convergence difficulties. Moreover, distillation may not effectively capture the temporal dynamics of video data, sometimes resulting in artifacts.

In contrast, our lower-bound guided training strategy (LBG) stabilizes the learning process by using a dual-VAE framework, which reduces pseudo-textures and ensures better temporal coherence. By leveraging the reference VAE as a contrastive signal, the difference between the output of the reference VAE and the lower-bound VAE is used to optimize the f16 VAE. We optimize the reconstruction quality while maintaining alignment with the real data distribution by LBG. This approach significantly outperforms both vanilla method and knowledge distillation, offering faster convergence, better training stability, and superior video quality in real-world scenarios. Our method also reduces the need for extensive hyperparameter tuning, providing a more efficient and robust solution for real-world video super-resolution tasks. Table 3b compares the performance of f16 VAE under different training strategies. Our lower-bound guided training strategy achieves the best performance.

## 5 CONCLUSION

We presented FastVSR, a codec-centric approach to diffusion-based Real-VSR that identifies the one-step bottleneck in the VAE and remedies it with an asymmetric design: a frozen f8 encoder preserves the pretrained latent interface while a high-compression f16 decoder performs indirect upsampling and concentrates HR computation into a single efficient pass. Coupled with a lower-bound–guided training scheme that supplies stable, probabilistically grounded supervision without adversarial instability, FastVSR achieves substantial end-to-end speedups—111.9× over multi-step diffusion and 3.92× over one-step baselines—while delivering competitive fidelity, strong perceptual quality, and robust temporal consistency at reduced compute and memory. We expect this codec-first perspective to generalize beyond super-resolution and, with extensions such as multi-scale asymmetric coding, lightweight transformer adaptation, and hardware-aware compression, to further narrow the gap between high quality and real-time deployment.

## ETHICS STATEMENT

The research conducted in the paper conforms, in every respect, with the ICLR Code of Ethics.

## REPRODUCIBILITY STATEMENT

We have provided implementation details in Sec.4.1. We will also release all the code and models.

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
