# SUPPLEMENTARY MATERIAL:
# ASYMMETRIC VAE FOR ONE-STEP VIDEO SUPER-RESOLUTION ACCELERATION

## 1 FREE ENERGY AND ITS APPLICATION IN VARIATIONAL AUTOENCODERS

Free energy, also known as the negative Evidence Lower Bound (ELBO), is a key concept in variational inference and plays an important role in the training of Variational Autoencoders (VAE). It quantifies the similarity between the approximate posterior distribution $q(z|y)$ and the true posterior $p(z|y)$, and guides the optimization of the model parameters. The free energy is defined as follows:

$$F(y) = \mathbb{E}_{q(z|y)}[-\log p(y|z)] + \text{KL}(q(z|y)\|p(z)), \tag{1}$$

where $y$ is the observed data, $z$ represents the latent variables, $p(y|z)$ is the likelihood of the data given the latent variables $z$, and $p(z)$ is the prior distribution over the latent variables. The term $\mathbb{E}_{q(z|y)}[-\log p(y|z)]$ represents the expected negative log-likelihood of the data under the approximate posterior $q(z|y)$, and the $\text{KL}(q(z|y)\|p(z))$ term measures the Kullback-Leibler (KL) divergence between the approximate posterior and the prior distribution.

The free energy serves as a lower bound on the log-likelihood of the observed data, and minimizing the free energy is equivalent to maximizing the Evidence Lower Bound (ELBO). The derivation of ELBO comes from the following inequality:

$$\log p(y) = \log \int p(y, z)dz = \log \int p(y|z)p(z)dz \geq \int q(z|y) \log \frac{p(y|z)p(z)}{q(z|y)}dz. \tag{2}$$

This lower bound provides a useful optimization target since, for complex models like VAE, directly computing the true log-likelihood $\log p(y)$ is challenging.

By minimizing the free energy $F(y)$, we effectively optimize the variational parameters in the VAE model, bringing the approximate posterior distribution closer to the true posterior distribution. This optimization process improves the quality of the learned latent space, enhancing the model's ability to generate high-quality data samples.

In the VAE framework, ELBO and free energy are central to the learning process. Free energy is a function of reconstruction error and the KL divergence, and during training, the model minimizes this objective, ultimately aligning the approximate posterior distribution with the true distribution. This optimization process stabilizes the training and enhances the model's ability to generate high-fidelity samples.

In practice, free energy is optimized using variational inference techniques, typically using stochastic gradient descent or other optimization methods. By optimizing the free energy, we align the approximate posterior distribution with the true distribution, improving the overall performance of the VAE model.

## 2 COMPUTE ANALYSIS AND FUTURE WORK

Figure 1 displays how the computational cost and inference time scale with resolution and the number of frames. As the resolution and frame count increase, both the compute cost and inference time grow, reflecting the increased complexity of the model's operations.

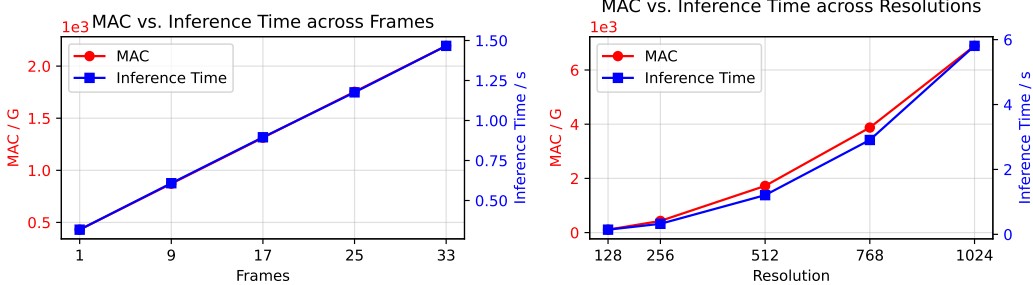

Figure 1: Scaling of MACs (G) and Inference Time (s) w.r.t. Frames and Resolution.

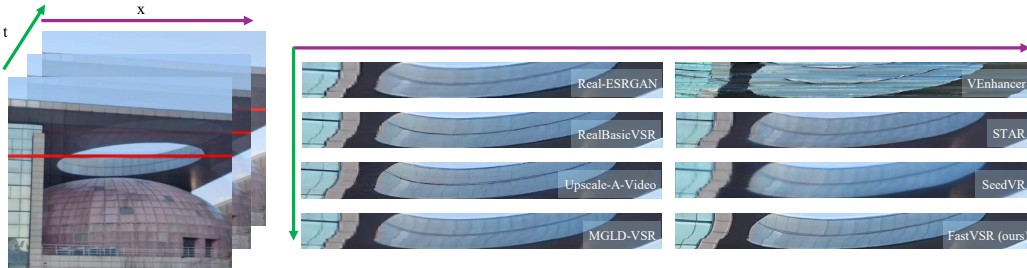

Figure 2: Comparison of temporal consistency (stacking the red line across frames).

In future work, we aim to explore the use of the F16-VAE with 8x time compression. This approach will focus on further reducing inference time while maintaining high-quality output, potentially enabling real-time deployment in more demanding applications.

## 3 COMPARISON OF TEMPORAL CONSISTENCY

The temporal profiles illustrated in Fig. 2 demonstrate that other methods exhibit noticeable line flickering, misalignment, and blurring across frames. In contrast, our method maintains smooth and stable temporal transitions, highlighting its superior temporal consistency.

## 4 MORE VISUAL COMPARISONS

In Figures 3 and 4, we provide additional visual comparisons to further highlight the clarity and realism of the results on UDM10 (Tao et al., 2017), SPMCS (Yi et al., 2019), YouHQ40 (Zhou et al., 2024), RealVSR (Yang et al., 2021), MVSR4x (Wang et al., 2023), and VideoLQ (Chan et al., 2022). These figures demonstrate the effectiveness of our approach in preserving high-frequency details while maintaining structural integrity across different types of degradation. The visual quality in terms of sharpness and texture realism is evident, showing that our method not only improves resolution but also enhances the perceptual accuracy of fine details.

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

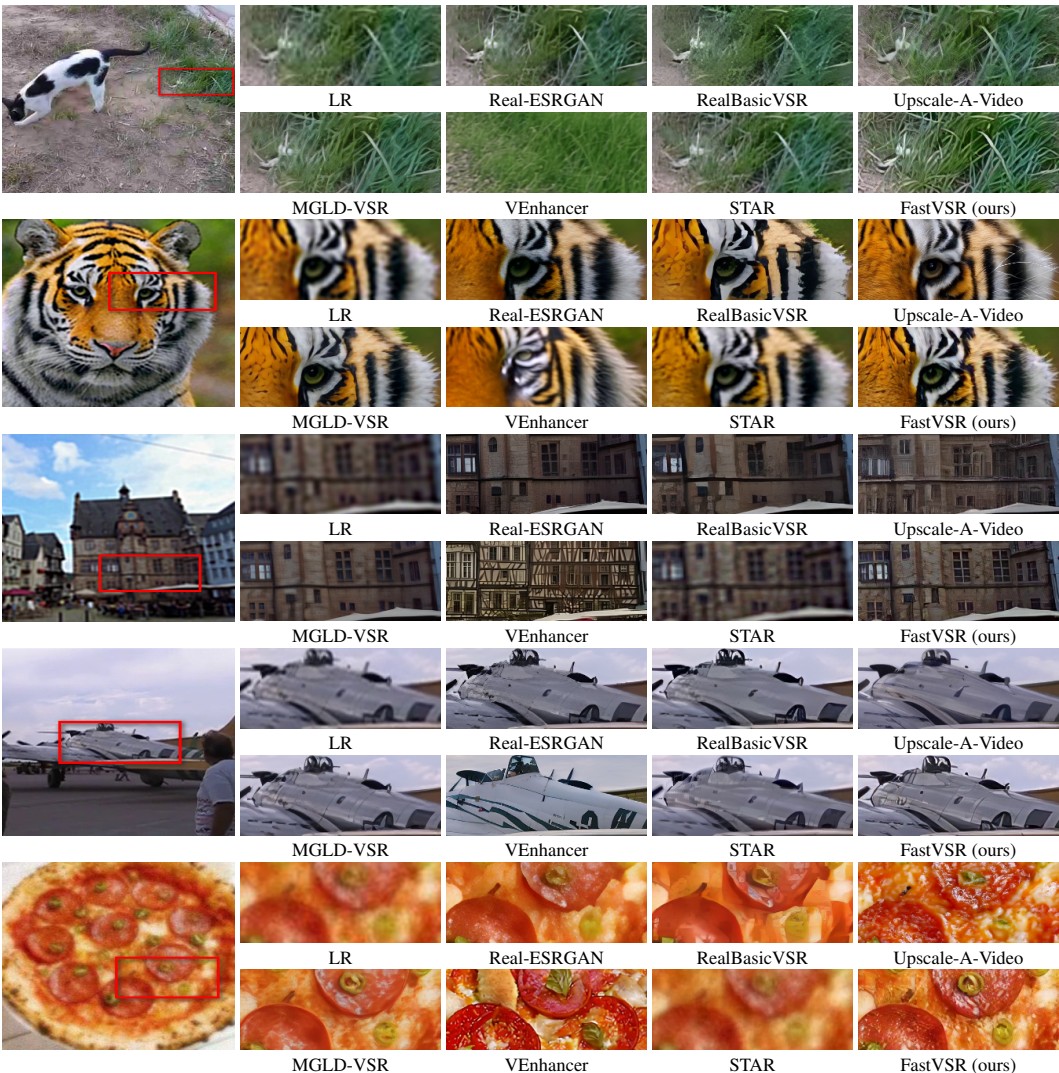

Figure 3: Visual comparison on synthetic datasets and real-world datasets.

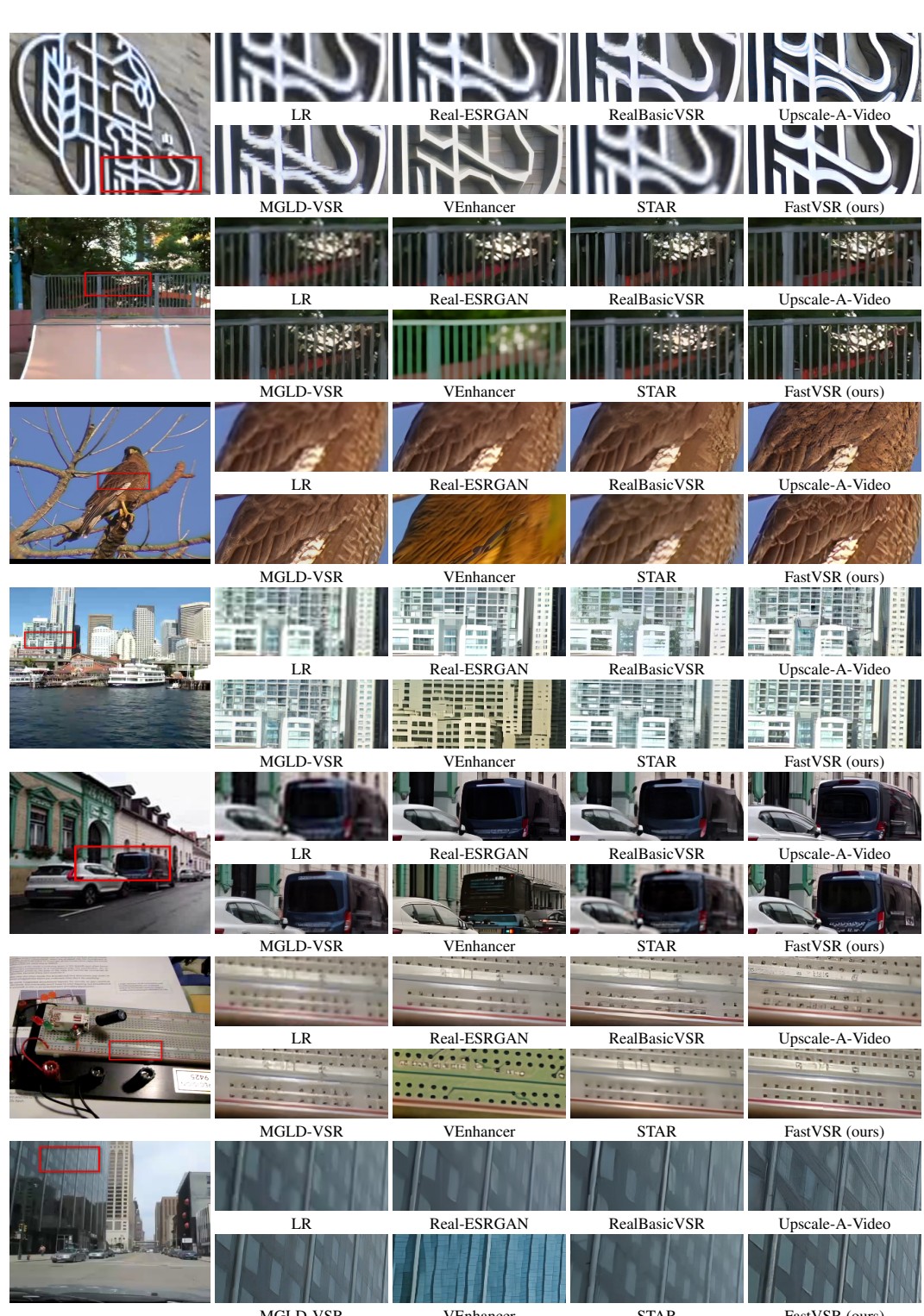

Figure 4: More visual comparison on synthetic datasets and real-world datasets.