# OpenReview forum: "Asymmetric VAE for One-Step Video Super-Resolution Acceleration"
_ICLR.cc/2026/Conference — ICLR 2026 Conference Withdrawn Submission_

### Official Review · Reviewer_esWS · 2025-10-26

**Soundness:** 2
**Presentation:** 2
**Contribution:** 1
**Rating:** 2
**Confidence:** 4

**Summary:**

The paper presents FastVSR, a diffusion-based video super-resolution model that improves inference efficiency. By introducing a highly compressed f16 VAE and a lower-bound-guided training strategy, the method reduces computation while maintaining stability and convergence. Using pixel shuffle and channel replication for upsampling, FastVSR achieves up to 111.9× faster inference than multi-step models and 3.92× faster than existing one-step models.

**Strengths:**

1. Propose to accelerate video super resolution by using a highly compressed f16 VAE.
2. Propose a lower-bound-guided training strategy to stable the training of their proposed f16 VAE.

**Weaknesses:**

1. Missing supplementary video. This is a video super-resolution paper, yet no supplementary video is provided. Without videos, it is difficult to evaluate the temporal consistency of the generated results.
2. Limited novelty. Increasing the compression ratio of the VAE is not a novel contribution. For example, LTX-Video already employs a 32×32×8 compression ratio. Moreover, LTX-Video highlights that excessively high compression may cause detail loss during decoding, and addresses this issue by jointly training the VAE with the denoise process to mitigate information loss. However, this paper does not provide any related analysis or justification.
3. Confusing details. In Lines 318–319, the authors mention that they initialize their model directly from the pretrained one-step model of CogVideo1.5. However, CogVideo1.5 itself is not a one-step model, which makes this statement unclear.
4. Incomplete comparison. The paper lacks comparisons with several strong state-of-the-art methods, such as DOVE [1] and RealViformer [2].
5. Inconsistent evaluation results. The reported performance of some baseline methods does not match their original papers. For example, SeedVR 3B/7B achieves 22.97/22.90 PSNR on the SPMCS dataset, which differs significantly from the 21.22 reported in Table 1.

[1] DOVE: Efficient One-Step Diffusion Model for Real-World Video Super-Resolution
[2] RealViformer: Investigating Attention for Real-World Video Super-Resolution

**Questions:**

1. What’s the difference between the proposed VAE and LTX-Video. What would be the outcome if the decoder from LTX-Video were directly used?
2. The paper only compresses the VAE, while the diffusion model itself—especially the attention modules—still contains substantial redundancy that could be further optimized. Why was this not explored?

---

### Official Review · Reviewer_WdKH · 2025-10-28

**Soundness:** 2
**Presentation:** 2
**Contribution:** 2
**Rating:** 2
**Confidence:** 4

**Summary:**

This paper introduces FastVSR, a one-step diffusion-based video super-resolution (VSR) framework that achieves speedups by improving the efficiency of VAE.
The key contributions include:

- An asymmetric VAE design — using an f8 encoder and a high-compression f16 decoder, allowing the diffusion transformer to operate in a more compact latent space while performing spatial upsampling only once at decoding.

- A lower-bound–guided (LBG) training strategy, which replaces adversarial losses with a dual-VAE free-energy objective that improves training stability and avoids pseudo-textures.

**Strengths:**

- The motivation is reasonable. Given the heavy design of existing VAE models for video generation, the efficiency of VAE becomes the bottleneck, especially for one-step diffusion VSR models. The authors provide a compelling computational analysis showing that, after moving to one-step diffusion, the VAE decoder dominates FLOPs and memory.

- The proposed dual-VAE lower-bound optimization shows a potential stable alternative for VAE finetuning.

- The paper is easy to follow, though with some vague details.

**Weaknesses:**

- The proposed approach is kind of incremental. A straightforward way of increasing the efficiency of VAE seems to be redesigning the architecture. Simply relying on an existing VAE via finetuning the decoder generally leads to trivial improvement, while the training difficulty could remain similar, i.e., both may face an unstable training process, while fixing the encoder may further share the same learning bias. Besides, a similar idea is also shared in previous work [1].

- If my understanding is correct, the main efficiency gain of the proposed method is due to the x2 upscale encoder, which leads to an x2 latent size rather than x4 compared with previous VSR baselines. The proposed method seems equal to conducting x2 upscaling first using a diffusion-based VSR and then using a GAN-based VSR (f16 VAE decoder) for another x2 upscaling. Such an operation generally leads to inferior performance compared with directly x4 upscaling using the diffusion-based VSR, though the efficiency can be improved, from my experience.

[1] Fine-structure Preserved Real-world Image Super-resolution via Transfer VAE Training. ICCV 2025

**Questions:**

My main concerns are as follows:

1. The major weaknesses above.

2. The training details of the proposed approach are kind of vague. From Lines 318, only the VAE decoder is finetuned, while the DiT backbone is from CogVideoX1.5, and is fixed during the training. To my knowledge, CogVideoX1.5 is not a VSR model. Then, it is confusing how the proposed approach could convert a T2V model to a VSR model by simply finetuning the VAE decoder?

---

### Official Review · Reviewer_QcZ9 · 2025-10-31

**Soundness:** 3
**Presentation:** 3
**Contribution:** 2
**Rating:** 4
**Confidence:** 5

**Summary:**

To reduce the dominant computational cost of the VAE codec in diffusion-based Real-VSR systems, this paper proposes a one-step video super-resolution framework built around an asymmetric VAE that performs indirect upsampling in the latent space and proposes a lower-bound-guided training strategy. It achieves significant speedup and memory savings while maintaining high perceptual fidelity and temporal consistency.

**Strengths:**

The authors propose a lower-bound-guided training strategy, which is demonstrated to be effective through ablation studies.
Extensive experiments demonstrate significant efficiency improvements (111.9× over multi-step, 3.92× over one-step) and 46.3% lower peak memory, with competitive perceptual and temporal metrics across multiple benchmarks.

**Weaknesses:**

1 The novelty is incremental. Many one-step diffusion models are proposed to solve the problems of visual tasks, what is special for the one-step for video SR.
2 The experimental results are not convincing, and in Tables 1 and 2, it is suggested to compare the proposed method with other one-step SR methods.

3 Section 3.1 argues that encoding and decoding between pixel and latent spaces are computationally expensive, with the decoding FLOPs being much higher than those of the denoising process. However, it is difficult to understand Equations (1)–(3), where many parameters are not introduced before used and their relationships are unclear. For instance, the definition and role of Act_max are not explained, and the meaning of μ in Equation (2) is ambiguous.
4 In the FastVSR architecture, for 4× super-resolution, the authors upsample 2× before processing and another 2× during decoding. I wonder what happens if directly performing a single 4× upsampling in the decoding stage and if it yields comparable performance with fewer FLOPs?
5 Figure 3 is not cited in the main text, and the training strategy described in Section 3.3 is difficult to align with the figure (e.g., the “Free Energy Difference” term shown in the figure). Moreover, certain implementation details are missing—are the parameters of the lower-bound VAE randomly initialized or derived from the Reference VAE? The meaning of H(·) in Equation (8) also requires clarification.
6 In Section 4.1, the statement “directly initialized from pretrained one-step model” lacks specificity. Please clarify whether this pretrained model was trained by the authors or taken from an existing one-step diffusion method. If it is the latter, its performance on the comparison datasets should be reported.
7 In Table 2, the term “f8 VAE one-step VSR”is not clearly defined. Please specify whether it refers to an existing method or a variant proposed by the authors, and include an appropriate citation or reference result.
8 A minor issue: does AdamW actually have a \beta_3 parameter? Typically, AdamW only includes \beta_1 and \beta_2.
9 Limited Exploration of the Core Asymmetric Design Principle: The paper's main architectural contribution—the asymmetric VAE—is only evaluated in a single configuration(f8-to-f16). The work would be substantially stronger if it included an ablation study on the asymmetry ratio itself. For instance, how does the framework perform with other configurations, such as f8-to-f32 or even f4-to-f16? It is unclear how training stability, computational cost, and reconstruction quality trade off as the asymmetry ratio changes. This lack of analysis leaves the reader wondering about the sensitivity and scalability of the proposed design choice.

**Questions:**

See Weakness

---

### Official Review · Reviewer_oF56 · 2025-11-01

**Soundness:** 3
**Presentation:** 3
**Contribution:** 3
**Rating:** 6
**Confidence:** 4

**Summary:**

This paper introduces FastVSR, a one-step diffusion-based video super-resolution (VSR) framework that accelerates inference by replacing the symmetric codec in conventional one-step diffusion models with an asymmetric variational autoencoder (VAE).
The encoder operates at 1/8 resolution (f8) and the decoder at 1/16 (f16), significantly reducing spatial token count.
A lower-bound-guided (LBG) training strategy, using a reference VAE and a lower-bound VAE, is proposed to stabilize learning without adversarial losses.
Experiments on REDS, RealVSR, MVSR4x, and other datasets show up to 111× inference speedup compared with diffusion-based multi-step VSR and a 3.9× gain over existing one-step models, with comparable visual quality.

**Strengths:**

**Practical and well-motivated innovation.**
   The asymmetric codec design addresses a real bottleneck — high decoding cost — with a clear quantitative motivation (Section 3.1).

 **Strong efficiency gains.**
   Experiments demonstrate impressive acceleration while maintaining perceptual quality, which is practically valuable for deployment.

 **Comprehensive experiments across datasets.**
   Evaluation covers synthetic and real benchmarks, demonstrating broad applicability.

 **Clear writing and strong empirical presentation.**
   The paper is well-organized, visually clear (especially Figure 2), and easy to follow.

**Weaknesses:**

**Shallow discussion of asymmetry impact.**
   The paper does not fully analyze *why* the f8–f16 asymmetry yields such a strong performance–efficiency trade-off.
   There is no quantitative study of reconstruction quality versus token count, or sensitivity to different encoder–decoder ratios.

 **No video visualizations.**
   Only static frames are presented. For a video task, lack of temporal visualizations or videos demonstrating motion stability weakens the perceptual claim.

In particular, the paper does not compare to DOVE: Efficient One-Step Diffusion Model for Real-World Video Super-Resolution (NeurIPS 2025), which is a directly relevant and contemporary baseline achieving strong performance-speed trade-offs.

**Questions:**

- What happens if both encoder and decoder operate at f16 (full symmetry)? Does quality degrade significantly?

- How does the lower-bound-guided training compare to simpler regularizers (e.g., KL annealing or latent noise)?

- Can this method generalize to different 1080p or 4K videos?

---

### Note · Authors · 2025-11-14

I have read and agree with the venue's withdrawal policy on behalf of myself and my co-authors.